

# Precipitable water vapor content from ESR/SKYNET Sun-sky radiometers: validation against GNSS/GPS and AERONET over three different sites in Europe.

Monica Campanelli[1], Alessandra Mascitelli[1,2], Paolo Sanò[1], Henri Diémoz[3], Victor Estellés[4], Stefano
Federico[1], Anna Maria Iannarelli[5], Francesca Fratarcangeli[2], Augusto Mazzoni[2], Eugenio Realini[6],
Mattia Crespi[2], Olivier Bock[7], Jose A. Martínez-Lozano[4], Stefano Dietrich[1]

[1]Institute of Atmospheric Sciences and Climate, (ISAC), National Research Council (CNR), Via Fosso del Cavaliere 100,
00133, Rome, Italy
[2]Geodesy and Geomatics Division – DICEA, University of Rome "La Sapienza", via Eudossiana, 18, 00184 Rome, Italy.
[3]Environmental Protection Agency (ARPA), Loc. Grande Charrière 44 - 11020 Saint-Christophe, Aosta, Italy.
[4]Dept. Física de la Terra i Termodinàmica, Universitat de València, Burjassot (Valencia), Spain.
[5]SERCO SPA - Via Sciadonna 24 - 00044 Frascati, Rome.
[6]Geomatics Research and Developments (GReD) srl, via Cavour, 2, 22074 Lomazzo (CO), Italy
[7]IGN LAREG, Univ Paris Diderot, Sorbonne Paris Cité, 5 rue Thomas Mann, 75205 Paris Cedex 13, France

*Correspondence to*: Monica Campanelli (m.campanelli@isac.cnr.it)

**Abstract.** The estimation of the precipitable water vapor content ($W$) with high temporal and spatial resolution is of great
interest in both meteorological and climatological studies. Several methodologies based on remote sensing techniques have
been recently developed, in order to obtain accurate and frequent measurements of this atmospheric parameter. Among them,
the relative low cost and easy deployment of sun-sky radiometers, or sun-photometers, operating in several international
networks, allowed the development of automatic estimations of $W$ from these instruments with high temporal resolution.
However the great problem of this methodology is the estimation of the sun-photometric calibration parameters. The
objective of this paper is to validate a new methodology based on the hypothesis that the calibration parameters
characterizing the atmospheric transmittance at 940 nm are dependent on vertical profiles of temperature, air pressure and
moisture typical of each measurement site. To obtain the calibration parameters some simultaneously seasonal measurements
of $W$, from independent sources, taken over a large range of solar zenith angle and covering a wide range of $W$, are needed.
In this work yearly GNSS/GPS dataset were used for obtaining a table of photometric calibration constants and the
methodology was applied and validated in three European ESR-SKYNET network sites, characterized by different
atmospheric and climatic conditions: Rome, Valencia and Aosta. Results were validated against the GNSS/GPS and AErosol
Robotic NETwork (AERONET) $W$ estimations. In both the validations the agreement was very high with a percentage
RMSD of about 6%, 13% and 8% in the case of GPS intercomparison at Rome, Aosta and Valencia, respectively, and of 8%
in the case of AERONET comparison in Valencia.





Analysing the results by $W$ classes, the present methodology was found to clearly improve $W$ estimation at low $W$ content when compared against AERONET in term of %Bias, bringing the agreement with the GPS (considered the reference one), from a %Bias of 5.76 to 0.52.

## 1 Introduction

The precipitable water vapor content, hereafter referred as $W$, is the total atmospheric water vapor contained in a vertical column of unit cross-sectional area extending between any two specified levels, commonly expressed in terms of the height to which that water substance would stand if completely condensed and collected in a vessel of the same unit cross-section (American Meteorological Society, 2015). The estimation of this quantity is of great interest in meteorological and climatological studies. Near real-time $W$ measurements can be used for weather diagnoses and forecasting applications (Poli

et al., 2007; Hong, 2015), and for studying meteorological disturbances occurring in some areas in order to improve the prediction of local heavy rainfall, a difficult task with current mesoscale numerical prediction models (Realini et al, 2014; Bock et al. 2016). Water vapor plays also a key role in the Earth's climate system leading the hydrological cycle and affecting the global radiation budget as greenhouse gas (Schmidt et al., 2010). Also microphysical processes leading to the formation of clouds are influenced by $W$ variations, whose effect on the size, shape, and chemical composition of aerosols

can modify their role in the direct and indirect radiative forcing (Yu et al., 2014; Haywood et al., 2011).
Therefore it is really important to perform measurements of $W$ with high temporal and spatial resolution, because of its high variability in both space and time across the Earth. Several methodologies have been recently developed in order to obtain accurate estimations: Global Navigation Satellite System (GNSS), including the Global Positioning System (GPS) (Bevis et al., 1992; Guerova et al., 2016), microwave radiometers (Elgered et al., 1982) and sun-photometers (Campanelli et al., 2014;

Halthore et al., 1997; Alexandrov et al. 2009), are the automatic remote sensing instruments able to provide precise and frequent measurements of $W$. All of them have some pro and cons. The GNSS methodology is based on the signal received from positioning satellites by ground-based antennas; $W$ calculated with this technique has a high temporal resolution (typically from 30 seconds to tens of minutes, depending on the processing strategy), and if the network of ground receivers is very dense, a high spatial resolution as well. However the networks of receivers do not provide automatically the final

product, that is W, and to get the highest accuracy out of the GPS measurements, careful post-processing using available scientific softwares must be performed (Bock and Doerflinger, 2001). For what concerns microwave radiometers, they have a very high temporal resolution (up to 1 second) and good accuracy but, because of the cost of the instrument and maintenance needed, there are not many installed all over the world. Finally, the relatively low-cost and easy deployment of sun-photometers, allowed the establishment of several international networks operating in the last 30 years worldwide.

AErosol RObotic NETwork (AERONET) network (Holben 1998; Smirnov 2004), Global Atmospheric Watching – Precision Filter Radiometer (GAW-PFR) network (Wehrli, 2000; Nyeki et al., 2005) and European Skynet Radiometers (ESR-SKYNET) network (Campanelli et al., 2012, 2014) are providing operationally free estimation of $W$, or developing





algorithms and open source packages for improving the retrieval of this important atmospheric parameter. However, the great problem of this methodology is the estimation of the sun-photometric calibration parameters. The objective of this paper is to validate a new methodology, originally developed and applied at a single site in Japan (Campanelli et al., 2014), at three European ESR-SKYNET sites characterized by different atmospheric and climatic conditions. The methodology is

based on the hypothesis that the calibration parameters characterizing the atmospheric transmittance at 940 nm (the wavelength used by sun photometers for retrieving $W$), are dependent on vertical profiles of temperature, air pressure and moisture typical of each measurement site. To obtain calibration parameters, simultaneous and season-dependent measurements of $W$ are needed, taken over a large range of solar zenith angle and covering a wide range of $W$. In the present paper yearly GPS datasets are used to obtain a table of photometric calibration constants for each site, covering a range of $W$

from 0 to 40 mm. Results are validated against a co-located AERONET sun photometer and a subset of GPS measurements.

## 2 Measurement sites and equipment

The methodology for the determination of precipitable water vapour content from sun-sky radiometers was applied to measurements taken at three ESR sites (Figure. 1): Rome (12.500° E, 41.900N°, 83 m asl, Italy), Burjassot-Valencia (0.418°W, 39.508°N, 60 m asl, Spain), and Saint-Christophe – Aosta (7.357° E, 45.742°N, 570 m asl, Italy), during three

different years: 2010, 2011 and 2014, respectively. The sites under study are different both in location and in atmospheric conditions.

Rome is an urban site, with about 3.0 million of inhabitants, 25 km east from the Mediterranean Sea, in the middle of an undulating plain. The atmosphere is affected by traffic emission, but also semi-rural particulates and, especially during summer season, by sea breeze and desert dust advection from Saharan region.

Burjassot is an urban site located in the metropolitan area of Valencia (Spain) with approximately 1.4 million inhabitants, 10 km west from the coast and 5 km northwest from the Valencia city center. Therefore, it is mainly affected by urban pollution, but also by rural aerosols from non-irrigated inland areas and marine aerosols from the Mediterranean Sea (mainly under the sea breeze dynamics). Occasional events of mineral dust from the Sahara and biomass burning from Mediterranean type forests happen mainly during summer months. In general, its environment is relatively humid, mainly when the site is

under the effect of the sea air mass; however, during the Saharan air advections, or when the wind flows mainly from inland Iberian Peninsula, the environment can be very hot and dry.

Saint-Christophe is located in the Aosta Valley, 3 km far from the small city of Aosta (about 30.000 inhabitants). The location is in a semi-rural context, partially influenced by anthropogenic activity both on local (emissions from cars, heating systems, steel mill) and regional (advection from the Po valley) scale. The prevalent wind circulation is forced by mountain-

valley and mountain-plain breezes, and Föhn episodes are not infrequent. The ground surface is covered with soil, grass, and buildings and during wintertime with snow.



The different atmospheric conditions typical of each site, and therefore the seasonal variability of $W$, make these locations appropriate for a validation of the proposed technique.

**2.1 Sun-sky radiometers**

A sun-sky radiometer is a narrow band filter photometer able to perform measurements of direct solar and diffuse sky irradiances at some selected wavelengths and at several scattering angles.

The three ESR sites under study are equipped with the standard SKYNET network instrument, that is a sun-sky radiometer model POM-01 or POM-02, produced by PREDE Co., Ltd, Japan. Although mainly used for studying atmospheric aerosol

optical and physical properties in clear sky conditions, these instruments also perform irradiance measurements at 940 nm for water vapour studies. The methodology used in this paper has been specifically developed for estimating columnar water content from POM sun-sky radiometers ($W_P$), but can also be applied to other spectral radiometers.

A CIMEL CE318 sun-sky radiometer part of AERONET network (often called simply sun photometer), is operating in Valencia since 2007, co-located with the ESR-POM since 2008. This instrument, also aimed at the characterization of

atmospheric aerosols, performs the same kind of measurements of POMs model, but has some technical differences that require a further calibration for the diffuse radiance, not needed in the POMs model (Campanelli et al., 2007). Precipitable water vapour content from this instrument ($W_{AER}$) is calculated using the official AERONET inversion algorithm (Smirnov et al., 2004) and successive updates, providing also the aerosol products.

Table I shows the most important characteristics of both the POM and CIMEL sun-sky radiometers.

**2.2 GNSS/GPS receivers**

Three dual-frequency GNSS/GPS receiver stations (hereafter called simply GPS) were used for estimating the precipitable water vapour content, hereafter called $W_{GPS}$.

The M0SE system (12.493° E,41.893° N, ellipsoidal height: 120.6m, altitude: 72.14m) is located on the roof of the Faculty of Engineering, University of Rome "La Sapienza", about 2km far from the sun-sky radiometer position. This system is part

of the EUREF Permanent GNSS Network and the observational files, in Receiver INdependent EXchange Format (RINEX) used for the retrieval of $W_{GPS}$, were provided by the Geodesy and Geomatics Division, University of Rome "La Sapienza".

The AOST system (7.345° E,45.741° N, ellipsoidal height: 624.1m, altitude: 570.97m), is located in Valle d'Aosta region, about half km far from the sun-sky radiometer position; it is part of the NetGEO network and the observational files, for the calculation of $W_{GPS}$, were provided by the NetGEO network operators. For both the above stations dataset with time interval

of 30 seconds was used to retrieve $W_{GPS}$ with a temporal resolution of 15 minutes.

The VALE system (0.3376°W, 39.481°N, ellipsoidal height:77.7m, altitude: 26.88m) is located on the roof of the Escuela de Cartografia y Geodesia of Universidad Politecnica de Valencia, Spain, about 7 km east from the sun-sky radiometer position.



This system part of the EUREF Permanent GNSS Network and RINEX data are available from EUREF website (http://www.epncb.oma.be). For this station $W_{GPS}$ was retrieved with time interval of 1 hour.

## 3 Methodology

### 3.1 GPS receivers

Water vapour content in the troposphere affects GNSS signals by lowering their propagation velocities with respect to vacuum. A diminished speed results in a time delay in the signal propagation along the satellite-receiver path, that multiplied by the vacuum speed of light adds an extra-distance to the satellite-receiver geometrical one. It is worth reminding here that the tropospheric delay (the word delay is usually referred to the extra distance and is expressed in meters) due to the water vapour, is just one out of many other systematic errors affecting GNSS observations, which are to be accounted for in order

to achieve sub-centimeter accuracy positions. If from the positioning point of view this delay is just a systematic error to be removed, it puts forward GNSS as a tool for the remote sensing of the troposphere water vapour content. Since many years, the meteorological community has started to consider this by-product of high-accuracy positioning as one of the available observations, and time series of GNSS tropospheric water vapour delays are currently assimilated by some number of numerical weather prediction models, in some cases in a routinely way. The methodology to retrieve columnar water vapour

content consists in the following steps:

1) the satellite-related parameters (ephemeris-orbits and satellite clocks) provided by the Centre for Orbit Determination in Europe (CODE) are used as input in the Bernese GNSS Software 5.0 for station VALE and 5.2 for stations M0SE and AOST (www.bernese.unibe.ch/docs/DOCU52.pdf), in order to estimate the values of ZTD (Zenith Total Delay) from the satellite-receiver range observations of a selected receiver. The Bernese software, developed at the Astronomical Institute of the

University of Bern (AIUB) is a scientific package meeting highest quality standards for geodetic and further GNSS applications. In this work, only the GPS signals were processed.
For what concerns the M0SE receiver, being part of the EUREF Permanent GNSS Network, the coordinates provided by CODE in the Solution INdependent EXchange format (SINEX) files are known and the Precise Point Positioning (PPP) absolute positioning and coordinates constrained methodology was used in the Bernese analysis. As regards AOST receiver,

the SINEX files are not available and the PPP and free network solution methodology was used, allowing the Bernese GNSS Software to estimate also the station coordinates. Station VALE was processed in double-difference mode within a global network as a loosely constrained solution (10 m). The ZTD estimates were quality checked based on inspection of ZTD time series and formal errors following the methodology described in Bock et al. (2016).

2) From the values of ZTD, $W_{GPS}$ can be calculated using Pressure and Temperature predicted by a NWP model or measured at the surface by weather stations located nearby the GNSS station. These dataset, were provided by ARPA Lazio for M0SE station and by Regione Valle d'Aosta for AOST receiver. For station VALE, the surface pressure values from ECMWF





reanalysis (ERA-Interim) were used. The height difference between receivers and weather stations has been corrected according to Realini et al. (2014). Latitude and altitude of receivers, read in the RINEX files are also used to convert ZTD to precipitable water content $W_{GPS}$, using the procedure by Bevis et al. (1992). This technique allows calculating the Zenith Hydrostatic Delay (ZHD) and the Zenith Wet Delay (ZWD), defined as the difference between ZTD and ZHD. Finally,

according to Askne et al., (1987), Eq.(1) returns the amount of IWV (Integrated Water Vapour),

$$IWV = K(Tm) * ZWD \tag{1}$$

where the coefficient K depends on the vertically integrated mean temperature (Tm) (Davis et al., 1985) and can be obtained

either from meteorological models or by the linear relationships proposed by Bevis et al. (1992): $Tm \sim 70.2 + 0.72 Ts$, where $Ts$ is the measured temperature. In the analysis of Aosta GPS data, the Bevis equation was used, whereas for VALE, ECMWF data were used. Consequently, the values of PWV are obtained dividing the value of IWV by the water density (1000 kg m-3). Many studies have assessed the accuracy of GPS IWV estimates by comparison with measurements from other sensors (e.g. microwave radiometers, radiosondes, lidars…). It is well recognized that results are dependent on the

IWV itself, and thus on the geographic location (cold/warm climates) and on time (cold/warm or dry/wet season), but also on GPS processing options and of course on the quality of measurements from the reference sensor. Recent measurement campaigns performed at mid-latitudes have demonstrated RMS differences < 0.1 cm or 4-7% (Bock et al., 2013; Bonafoni et al., 2013; Pérez-Ramirez et al., 2014) which can be considered as representative for this study.

**3.2 Sun sky radiometers**

**3.2.1 ESR/PREDE-POM**

Precipitable water vapour content from ESR/PREDE-POM sun-sky radiometer ($W_P$) was calculated using the methodology described in Campanelli et al. (2014). The main aspect of this technique is the consideration that the atmospheric transmittance in the water vapour band, $T = e^{-a(mW)^b}$, with $m$ the optical airmass and $W$ the columnar water vapour content (Bruegge et al., 1992), depends on the vertical profile of temperature, pressure and moisture of each site of measurement, as

much as its characteristic parameters $a$ and $b$, whose values depend on the characteristics of the interferential filter, but also vary with the columnar water vapour amount. This procedure allows the estimation of $a$ and $b$ parameters directly from the measurements taken by the sun-sky radiometer, potentially containing the information on seasonal changes in vertical profiles of temperature, air pressure and moisture occurring in the site of measurement, and not relying on any radiative transfer calculation, therefore reducing simulation errors.

The direct solar irradiance $F$ (mA) measured by the POM sun-sky radiometer at the 940 nm wavelength in clear sky conditions, can be expressed by Eq. (2):

$$V = V_0 e^{-m(\tau_a + \tau_R)} e^{-a(mW)^b}, \tag{2}$$



where $V_0$ is the solar calibration constant, that is the extra-terrestrial solar irradiance in current units (mA); $m$ is the relative optical airmass (Kasten and Young, 1989); $\tau_a$ and $\tau_R$ are the extinction aerosol optical depth and the molecular Rayleigh scattering at 940 nm respectively; $T = e^{-a(mW)^b}$ is the atmospheric transmittance at 940 nm.

The procedure for the retrieval of $a$, $b$ and $V_0$, completely described in Campanelli et al. (2014), is here briefly summarized.

Equation (2) can be also written in the form,

$$y = lnV_0 - ax, \tag{3}$$

with $\quad \begin{cases} y = lnV + m \cdot (\tau_a + \tau_R) \\ \quad\quad x = (m \cdot W)^b \end{cases}$ . $\tag{4}$

$\tau_a$ is estimated at wavelength $\lambda$ =940 nm, according to the well-known Ångström formula in Eq.(5),

$\tau_a(\lambda) = \beta\lambda^{-\alpha} \tag{5}$

where $\alpha$ is the Ångström exponent, and $\beta$ is the atmospheric turbidity parameter. $\alpha$ and $\beta$ are determined by the regression from Eq. (5) where the spectral series of $\tau_a$ are retrieved by the sun-sky radiometer measurements taken at the other visible and near infrared wavelengths 400, 500, 675, 870, and 1020 nm.

From Eq.(4) $x$-values are calculated for several different values of $b$ and each time the $(x, y)$ squared correlation coefficient is

calculated; then the maximization of the $(x, y)$ squared correlation coefficient is used to determine the best exponent $b$. Once the optimal $b$ is retrieved, the series of $x$-values is computed and used in Eq. (3) where the regression line of $y$ versus $x$ allows the retrieval of the coefficients $a$ and $V_0$. The errors affecting $a$, $b$ and $V_0$ retrievals are evaluated using a Monte Carlo method as explained in (Campanelli et al., 2014).

This regression line is a modified version of Langley plot where $V_0$ is retrieved by plotting $y$ versus the product $ax$, with

$x = (m \cdot W)^b$. This approach, as demonstrated in Campanelli et al. (2014), extends the application of the Langley methods to cases where the time patterns of $W$ is not stable.

Once parameters $V_0$, $a$ and $b$ have been determined, the values of precipitable water content $W_P$ can be calculated according to the Eq.(6):

$$W_P = \frac{1}{m} \cdot \left[ \frac{1}{a} \cdot (lnV_0 - y) \right]^{\frac{1}{b}}. \tag{6}$$

In order to calculate $x$ values in Eq. (4), an independent dataset of columnar water vapour $W$, measured by other instrumentation (such as radiosondes, microwave radiometers or GPS receivers) taken over a large range of solar zenith angle, simultaneously with the sun-sky radiometer irradiance measurements, is needed. If seasonal dependent measurements of W are available, it is possible to calculate a table of calibration constants $(a, b)$ as function of the amount of columnar water vapour typical of the site under consideration. This table can be used for the calculation of $W_P$ until the instrument is

moved to another location or its status is deteriorated.





### 3.2.2 AERONET/CIMEL

In the AERONET methodology (Perez-Ramirez et al., 2014), the transmittance T at 940 nm is fitted to those generated from the HITRAN 2000 spectral database (Rothman et al., 2003) using the Spherical Harmonics (SHARM) radiative transfer code (Lyapustin, 2005), then the coefficients $a$ and $b$ are computed by a curve-fitting procedure of $T$ as a function of $W$. The output of the HITRAN 2000 spectral database is convolved with recently measured filter response functions (Smirnov et al., 2004). Each AERONET instrument has its own unique set of $a$ and $b$ values depending on the filter configuration, and these coefficients are considered fixed until the filter is changed. Since only one pair of $a$ and $b$ parameters is used, the dependence of $T$ on the vertical profile of temperature, pressure and moisture that can seasonally happen at each site, is neglected, introducing uncertainties in their retrieval. The use of a different database for the determination of water vapour transmittance, could also affect their value.

Once the coefficients $a$ and $b$ are known, the calibration constant $V_0$ is calculated by another modified Langley method, from observations taken at a high mountain site (Reagan et al., 1986; Bruegge et al., 1992; Halthore et al., 1997). This modified method, differently from the one used in the ESR/PREDE-POM procedure, determines $V_0$ as the intercept of the straight line obtained by fitting $y$ versus the power term $m^b$ in Eq (3 and 4). In this case, as demonstrated in Campanelli et al., (2014) the stability of the $W$ time pattern is required in order to avoid calibration errors. Finally using the retrieved calibration parameters $a$, $b$ and $V_0$ the precipitable water vapour content, $W_{AER}$, can be calculated from Eq (6).

### 4 Estimation of calibration constants

$W_{GPS}$ obtained by the three GPS receivers in Rome, Aosta and Valencia was used as independent dataset for calculating the calibration constants of the co-located ESR/PREDE-POM sun-sky radiometers. The cloud screening of radiometers measurements was performed by selecting those measurements whose RMSD between measured and reconstructed diffuse sky irradiance at all the wavelengths, used for aerosol study, and all angles, is lower than 20%. For the Rome site, an additional procedure was applied consisting in inter-comparing the selected measurements with those provided by a co-located Multi Filter Rotating Shadow band Radiometer (MFRSR), whose cloud screening is performed following the methodology by Alexandrov et al. (2004).

The closest $W_{GPS}$ retrievals within 15 min before and after the sun-sky radiometer measurements were selected. Then the simultaneous $[W_{GPS}, V]$ dataset was divided in two parts by picking every other day, among the available days: one part $[W_{GPS1}, V_1]$ was used for the application of the methodology and then the estimation of the calibration constants, the other part of GPS estimations ($W_{GPS2}$) was used to validate the $W_P$ retrievals. The two GPS dataset were found being equally populated and with similar frequency distributions, and the statistical independence between the $W_{GPS2}$ data used for the validation and $W_P$ was ensured.

Because $a$ and $b$ parameters are supposed to depend on the total amount of water vapour, the entire yearly independent $W_{GPS1}$ dataset was divided in 3 classes: [0 – 10] mm; [10 – 20] mm and [20 – 40] nm; an insufficient number of points was found





with water vapour larger than 40 mm, for the 3 sites. The sun-sky radiometer calibration parameters ($a$, $b$, and $V_0$) for each site and class were calculated (Table II, Figure 2).

In the first application of this methodology (Campanelli et al. 2014), performed for calibrating a sun sky radiometer at a Japanese site characterized by a wide yearly range of $W$ (from few mm up to about 60 mm), nearly parabolic opposite

behaviors of $a$ and $b$ as function of $W$ were found. The similar behavior of the boundary $W$ classes (being two maxima of the distribution) was demonstrated being linked to two different atmospheric regimes, with similar $W$ vertical distribution: trapping of $W$ due to winter inversion and occurrence of convection in summer. Both these regimes have a vertical structure with a well-mixed layer at the bottom and a rapid decrease upward. This behavior is recognizable in Aosta for the higher $W$ class. In this season in fact the well-mixed layer at the bottom is likely due to humid polluted air masses transported from the

Po Valley region, starting from late afternoon and staying in the atmosphere up to the morning. This advection was observed by ceilometer measurements for what concern the increase of suspended particles in the atmosphere, and by hygrometers for the growth of absolute humidity (Díemoz et al., 2016). Unfortunately there are no vertical profile measurements of $W$ in this site to verify this statement. Conversely in Rome and Valencia all the classes seem characterized by similar synoptic situations.

Looking at Figure 2, a slight fictitious tendency of $V_0$ on the water vapour class is recognizable. We remind that the retrieved $V_0$ in this methodology should be considered as an effective calibration constant whose variation could not be related to a real instrumental drift. Nevertheless, its total uncertainty (estimated as the standard deviation of the assumed values in each class divided by their average) resulted to be about 4%, 8% and 14% for Rome, Valencia and Aosta, respectively.

The uncertainty affecting the retrieval of $W_P$ ($\Delta W_P$ %) was estimated (as in Eq 7) by calculating the percentage RMSD

between $W_P$ and the $W_{GPS1}$ dataset, used for calibrating the sun sky radiometer:

$$RMSD = \sqrt{< (W_P - W_{GPS1})^2 >};$$

$$\Delta W_P \% = \frac{RMSD}{<W_{GPS1}>} \cdot 100; \tag{7}$$

The estimated uncertainties (Table II) values are comparable with that of AERONET retrievals (Perez-Ramirez et al., 2014) that is approximately 10%, with the exception of Aosta where a value of 20%, with a RMSD of 2.7 mm, is obtained. This is due mainly to two reasons: one is that the denominator $<W_{GPS1}>$ in Eq 7 is smaller for Aosta than for the other two sites,

resulting in higher $\Delta W_P$ % value; the other is related to the performance of GPS measurements in sites with rough orography. In fact, the methodology used for the calculation of ZTD assumes an azimuthal isotropy of the atmosphere above the antenna, within a conical field of view with an angular aperture of about 170° (since the elevation angle cut-off was set to 5°) centered in the site where the antenna is located. However, the orography can make the distribution of fluxes at high level quite complex and not uniform; the rougher the orography, such that surrounding the Aosta site, the greater is the

atmospheric anisotropy and therefore, in principle, the error introduced by the failure of the hypothesis assumed by GPS methodology. This degradation of the quality of GPS retrieval, not quantifiable, together with the missing of a large amount





of GPS data in Aosta during the summer months, make the fitting procedures used for the retrieval of calibration constants less stable and therefore increased the uncertainty in their estimation.

## 5 Intercomparison of methodologies

Once the calibration parameters $a,b$ and $V_0$, characterizing each sun-sky radiometer, were estimated for all the water vapour

classes, water vapour from sun-sky radiometer $W_P$ was directly calculated as in Eq (6) using the Table II parameters and the iterative procedure described in Campanelli et al., (2014). Figure 3 shows the retrieved time pattern of $W_P$ for all the sites. As expected winter season is the driest period in the three sites, and in summer Aosta shows a lover $W$ content compared to Rome and Valencia. Seasonal angle histogram of the hourly distribution of $W$ values grouped according to their numeric range*,* were performed in order to highlight the main differences among the 3 sites. Looking at Figure 4, referred to

summertime, it is worth highlighting that Valencia is the site where high $W$ values (>30 mm) are more homogenously distributed over time, with a very slight increment in the afternoon due to breeze circulation . This is principally due to the location of this site, very close to the sea, from where humid airmasses are transported all over the day. This kind of distribution of greater water vapor content is visible also in the other seasons, showing a sort of homogeneity of $W$ distribution all over the year. In Rome $W$ values >35 mm are mostly recognizable during summer afternoons, from about 14

UTC, due to the presence of a breeze circulation, advecting air from the sea. In all seasons, however, greater water vapor content is retrieved in the early morning and late afternoon showing, also for this site, a generally homogeneous $W$ yearly distribution. A smaller number of measurements is available in Rome during the middle par of the day in all seasons. This is mostly due to the formation of convective clouds at around 12 UTC, favored by the urban heat island phenomenon, that didn't allow the photometer to operate. In Aosta elevated values of $W$ (>35 mm) during summer were retrieved more

frequently in the morning, but this hourly distribution was found also in autumn for $W$>25 mm. This behavior could be caused by the atmospheric stability; in the late morning, especially in summer and fall when the insolation is higher, valley-mountain flows develop mixing the humid air of the lower levels with the dried air above. Then, winds aloft could remove part of this humidity by advection, decreasing the water content of the air column. However, further analyses, as the correlation between the humidity and the wind, are necessary to confirm this point. The other seasons conversely show more

homogeneous $W$ distribution during the day.

$W_P$ for each site was then validated against $W_{GPS2}$ (the part of the GPS dataset not used for the calibration) comparing measurements within 1 minute of difference. RMSD and Bias, defined in Eq 8, as well as squared correlation coefficient $R^2$, slope and intercept of the fitting straight line, were used for the statistical analysis, whose results are shown in Table III and Figure 5.

$$RMSD = \sqrt{<(W_{GPS2}-W_P)^2>}; \quad \%RMSD = \frac{\sqrt{<(W_{GPS2}-W_P)^2>}}{<W_P>} * 100;$$





$$Bias = <(W_{GPS2} - W_P)>; \qquad \%Bias = <\left(\frac{W_{GPS2} - W_P}{W_P} * 100\right)>; \tag{8}$$

The comparison between $W_P$ and $W_{GPS2}$ for Rome and Valencia (Table III) shows high $R^2$ when all $W$ classes are analyzed, with values of 0.98 and 0.96, %RMSD of 6.43% and 8.09%, and %Bias of -0.05% and 0.34% (therefore within the estimated error $\Delta W_P$), respectively. Investigating separately the 3 classes (divided using the thresholds on the $W_{GPS2}$ dataset) the

greatest difference was found for the first class in terms of %RMSD (9.18% and 14.51%) whereas the %Bias remained for each class within the $\Delta W_P$ error.

The retrieval of $W_P$ for Aosta was generally less performing than for the other sites. For the entire $W$ classes, %RMSD and %Bias were found to be the highest values, being 13.57% and -3.45% respectively, while $R^2$ is the lowest among the three sites (0.95). Also for this site the greatest value of %RMSD was found for the first class (18.00%) and the %Bias remained

for each class within the $\Delta W_P$ error. The lower quality performance of the methodology in this site is discussed in section 4.

Co-located with the ESR-SKYNET/POM, by the University of Valencia, there is an AERONET/CIMEL simultaneously operating in 2011. Since the philosophy used by the two networks for the radiometer's calibration is substantially different, it is worthwhile comparing the columnar precipitable water content estimated by the two methodologies and verify if there are

some improvements in assuming the calibration parameters dependent on the vertical distribution of $W$, and then on its total amount, respect to the commonly used assumption of fixed values.

Figure 6 shows the scatter plot of AERONET estimation ($W_{AER}$) vs $W_P$ for measurements within 1 minute. A very high squared correlation coefficient was found (Table IV) between the two series (0.99), with a total %Bias of 1.61%. However, analysing the results by classes, a larger discrepancy is evident for the first class in terms of %RMSD (10.33%) and %Bias

(9.16%) within the combined uncertainty of both $W_P$ and $W_{AER}$, with an overestimation of AERONET retrieval.

$W_{AER}$ was then compared against $W_{GPS}$ showing a %Bias of -0.97% and %RMSD of 7.62%, if all the classes are considered. The negative bias, consisting in an underestimation of AERONET retrieval, was also documented by Pérez-Ramirez et al., (2014) who found values varying from 2.2% to 7.9% depending on the site under study, when compared estimations from AERONET versus GPS. However the agreement against GPS was not so good for the first class of $W_{AER}$ where a higher

positive discrepancy was found (%Bias of 5.76) with respect to the one showed in the comparison between $W_P$ and $W_{GPS}$ (0.52%) in the same class. Looking at Figure 7, in fact, it is clear that the difference between $W_{AER}$ and $W_P$ has a decreasing trend with increasing $W$ (Figure 7, a) being greater than 10% (threshold of AERONET uncertainty) mostly for values below 15 mm, which is about the first class of $W$. A similar trend (even if less marked) is visible between $W_{AER}$ and $W_{GPS}$ (Figure 7, b). Conversely there is no clear tendency in the difference between $W_P$ and $W_{GPS}$ (Figure 7, c). This confirms that the

methodology here proposed takes to a general improvement of $W$ estimation, particularly evident for low $W$ content, in agreement with the findings in Campanelli et al., (2014), where the assumption of variable of $a$ and $b$ parameters increased the agreement with GPS retrievals of about 10% for the first $W$ class respect to the commonly assumption of fixed $a$ and $b$.



## 6 Conclusions

A methodology for retrieving precipitable water content $W_P$ from sun-sky radiometers measurements at 940 nm was applied to three sites of the ESR-SKYNET network with different atmospheric and climatic conditions. In order to provide $W_P$, the sun-sky radiometer must be calibrated in terms of the solar calibration constant $V_0$, that is the solar radiation incident at the

top of the atmosphere, and the calibration parameters characterizing the atmospheric transmittance at this wavelength, $a$ and $b$. This methodology considers that $a$ and $b$ are dependent on vertical profiles of temperature, air pressure and moisture typical of each measurement site, and therefore allows for the calculation of pairs of $(a, b)$ values for several classes of $W$. To obtain the calibration parameters, GPS-based yearly independent estimations of $W$, simultaneously with the sun–sky radiometer measurements, were used and a table of calibration constants, covering a range of $W$ from 0 to 40 mm, was built

for each site. In this work the GPS dataset was divided in two parts by picking every other day, among the available days: one part was used for the calibration of the sun-sky radiometer, the other part for the validation.

In principle it is needed to use one entire year of $W$ independent measurements for building the calibration table, however also a smaller dataset can be used, provided that it is taken over a large range of solar zenith angle and it covers a wide range of $W$.

The obtained $W_P$ values were characterized by an uncertainty $\Delta W_P$ below 10% for Rome and Valencia, and of 20% for Aosta. The yearly time pattern of $W_P$ for each site was then validated against the part of the GPS dataset not used for the calibration and against an AERONET sun photometer co-located in Valencia. In the former case for Rome and Valencia the agreement was found to be within the uncertainty $\Delta W_P$ when all the classes together are analyzed, whereas for Aosta a %RMSD of about 14% was found. Investigating separately the 3 classes, the greatest difference was found for the first class

in terms of %RMSD: 9.18%, 14.51% and 18%, for Rome, Valencia and Aosta respectively.

When compared against the AERONET retrieval, the agreement was found very good and within the uncertainties of both methodologies, if all the classes together are considered. However analyzing the results by classes, and after a cross-check among $W_P$, GPS and AERONET estimates, it was highlighted that the present methodology is able to generally improve $W$ estimation, particularly for low $W$ content in term of %Bias, bringing the agreement with the GPS (considered the reference),

from a %Bias of 5.76 to 0.52. This finding is in agreement with what already demonstrated in Campanelli et al., (2014), where the assumption of variable of $a$ and $b$ parameters was compared with the results from the assumption of fixed $a$ and $b$.

The present methodology can be easily applied to other kind of sun photometers or radiometers measuring the solar direct radiation at 940 nm wavelength, as Precision Filter Radiometers (PFR) or Multi Filter Rotating Shadowband Radiometers (MFRSR), provided that Angström, exponent and aerosol optical depth at 940 nm are available. The calibration table

containing $a$ and $b$ values for each $W$ class, can be used until the instrument is not moved to another location, or submitted to maintenance. In these cases all the calibration parameters must be recalculated.

The problem in the application of this methodology is however the availability of an independent, simultaneous $W$ dataset to be used for calibrating the sun-sky radiometer, or any other similar sun photometer. In Campanelli et al., (2014) the





possibility of using a rough estimation of the needed *W* dataset was tested, using surface observations of moisture parameters that are much more common than *W* values estimated by other equipment. The test provided very interesting results but still needs to be improved and validated in different sites and climatic conditions, that will be the next task for this kind of research.

5 **Acknowledgments**

We thank ARPA Valle D'Aosta and ARPA Lazio for providing ground measurements observations needed for the elaboration of GPS measurements.

We also thank Dr. Marco Cacciani of the University of Rome, La Sapienza, Department of Physics, G24 laboratory, for hosting the PREDE-POM sun-sky radiometer of Rome.

10 The Spanish Ministry of Economy and Competitiveness and the Valencia Autonomous Government are acknowledged for projects CGL2015-64785-R, CGL2015-70432-R and PROMETEUII/2014/058, that allowed the Burjassot-Valencia site measurements.

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



**Figures**

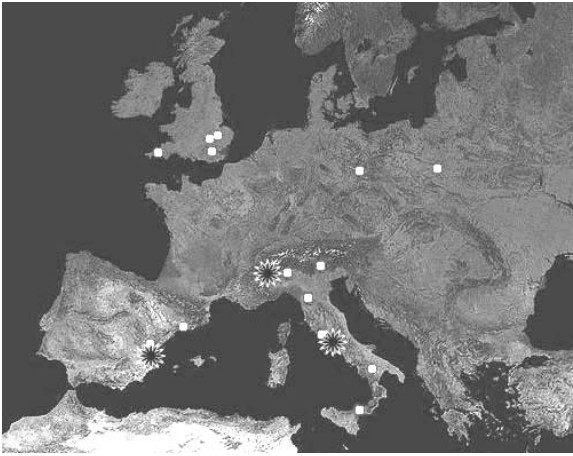

**Figure 1: Geographical position of ESR sites in the European region (white dots) and of the stations used in this work (black stars).**



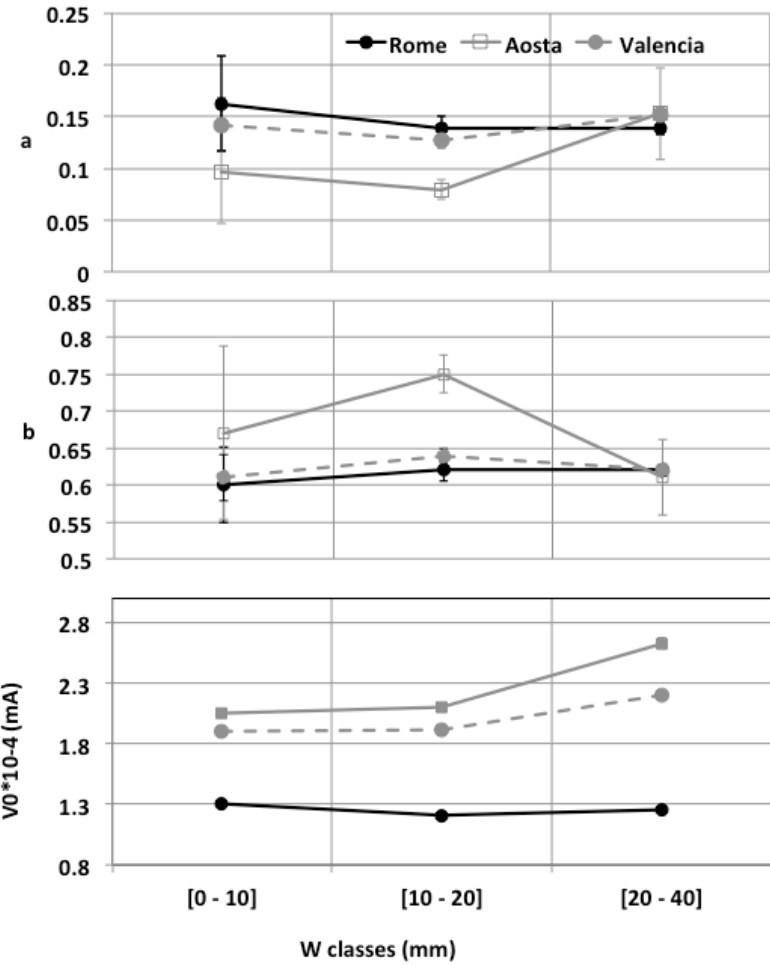

**Figure 2: behaviour of the estimated calibration parameters vs _W_ classes. The errors bars are the errors affecting the parameters as evaluated using a Monte Carlo method.**





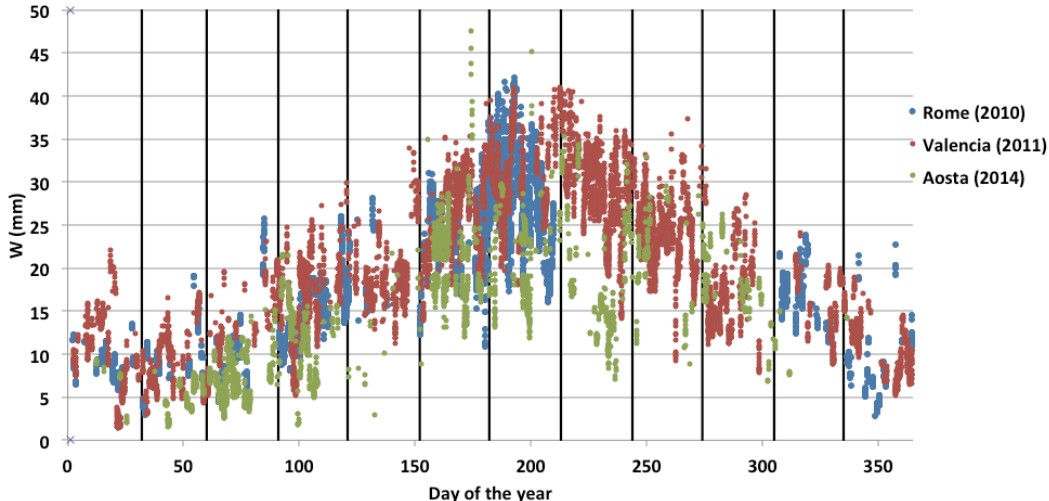

**Figure 3: time pattern of $W_P$ for the three sites under study**





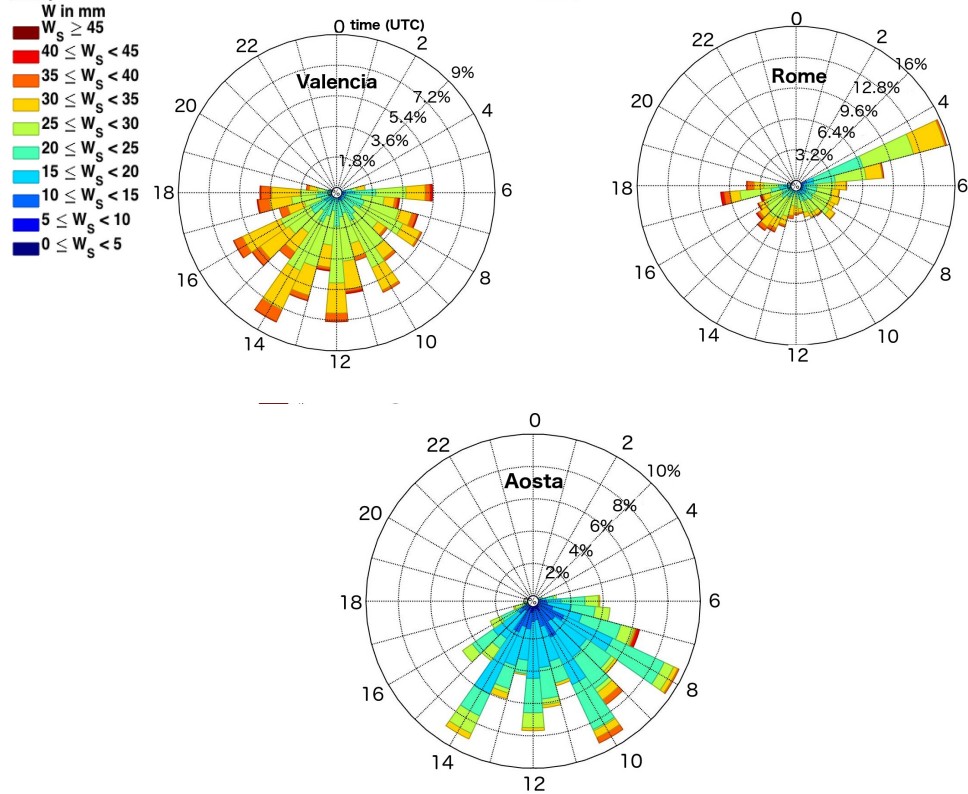

5    **Figure 4: polar plot showing the distribution of W values grouped according to their numeric range; the 24 quadrants are hours in**
     **UTC; the radius represents the frequency of events normalized to the number of point of the season.**





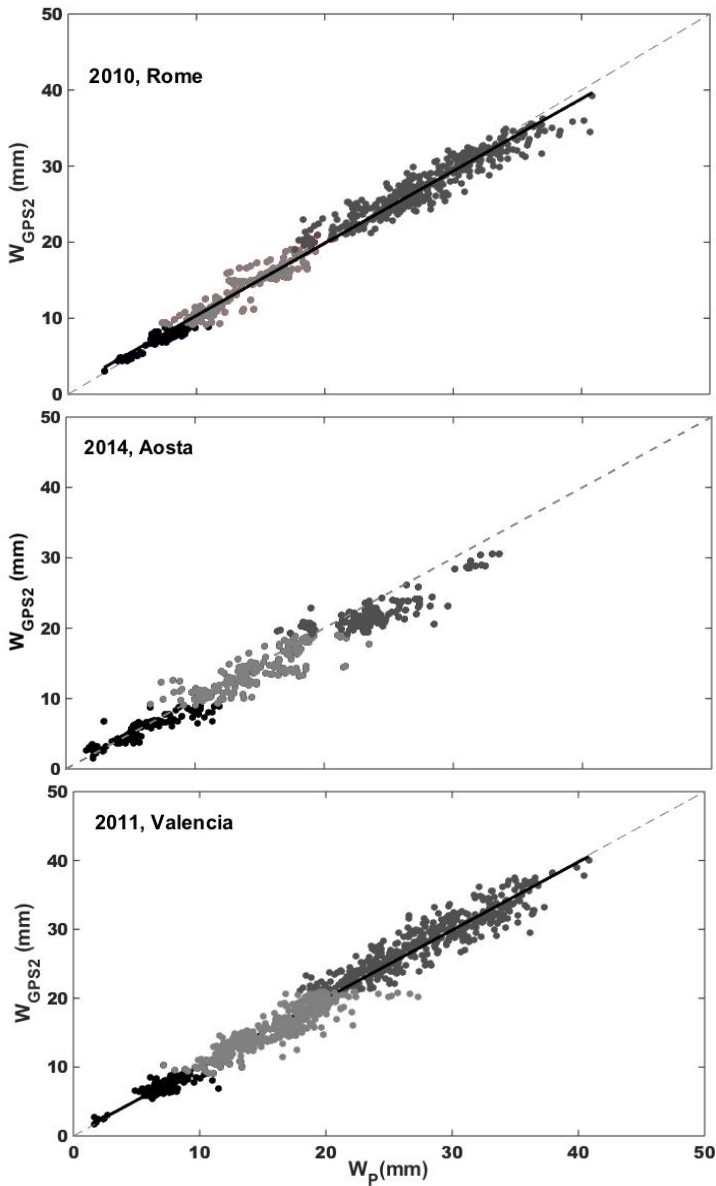

Figure 5: Scatter plot of $W_{GPS2}$ vs $W_P$. Alternations of black and greys indicate the three water vapour classes



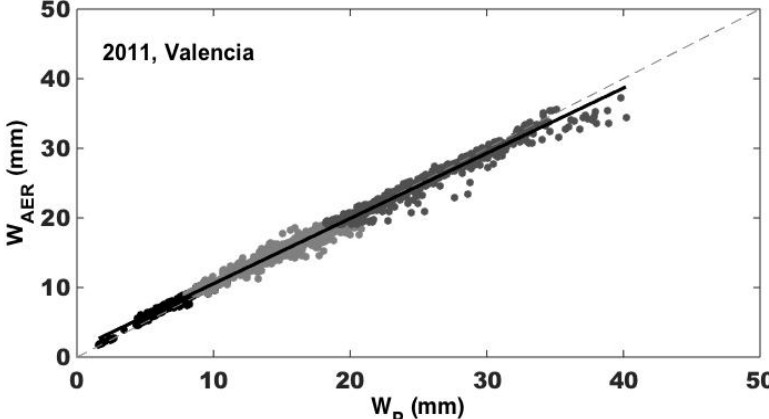

Figure 6: Scatter plot of $W_{AER}$ vs $W_P$; Alternations of black and greys indicate the three water vapour classes

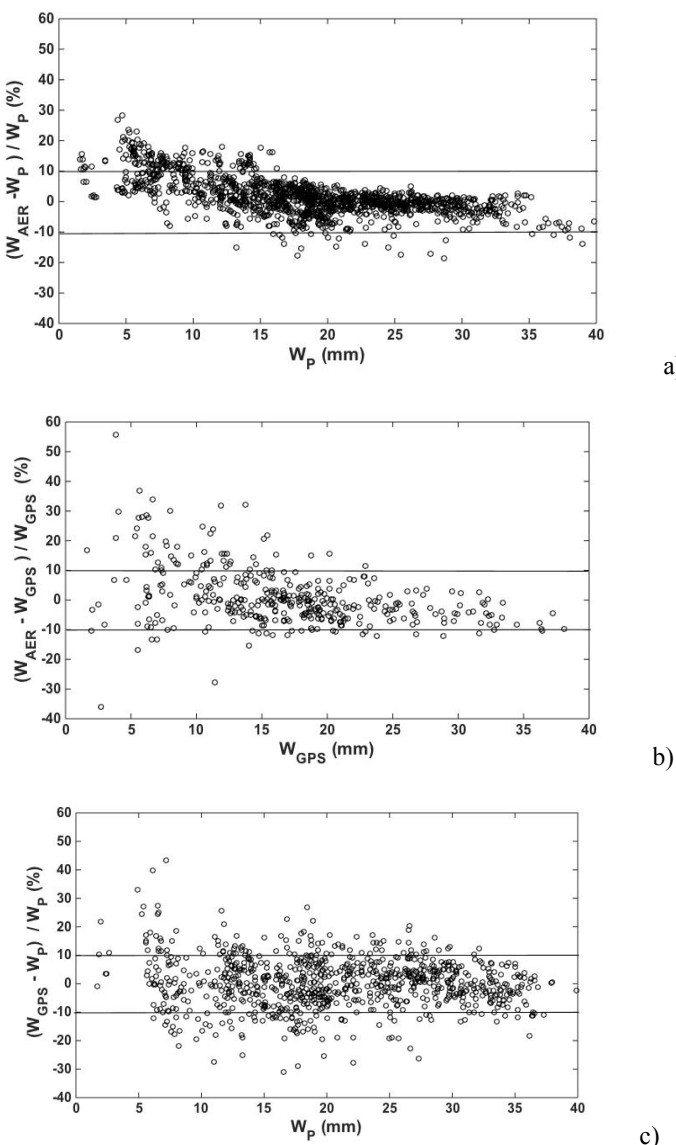

Figure 7: percentage relative differences between $W_{AER}$ and $W_P$ (a), $W_{AER}$ and $W_{GPS}$ (b), and $W_{GPS}$ and $W_P$ (c)



| Site | Network | Model | Wavelengths | View Angle |
|------|---------|-------|-------------|------------|
| Rome | ESR | PREDE-POM01 | 340,400,500, 675,870,940,1020 | 1.0° |
| Aosta | ESR | PREDE-POM02 | 315,340,380,400,500,675,870, 940,1020,1600,2200 | 1.0° |
| Valencia | ESR | PREDE-POM01 | 315,440,500,675,870,940,1020 | 1.0° |
| Valencia | AERONET | CIMEL-CE318 | 340,380,440,500,675,870,940, 1020,1640 | 1.2° |

**Table I: characteristics of the sun-sky radiometers located at the three sites. The exact wavelengths of the CIMEL at Valencia can slightly change depending on particular instrument replacements**





| Classes (mm) | N. points | $a$ | $b$ | $V_0$ x $10^{-4}$ (mA) | $\Delta a$ | $\Delta b$ | $\Delta V_0$ x $10^{-4}$ (mA) | $<W_{GPS1}>$ (mm) | RMSD (mm) | $\Delta W_P$ % |
|---|---|---|---|---|---|---|---|---|---|---|
| **Rome** | | | | | | | | | | |
| [0 – 10] | 29 | 0.162 | 0.60 | 1.31 | 0.046 | 0.05 | 0.01 | | | |
| [10 – 20] | 162 | 0.138 | 0.62 | 1.21 | 0.012 | 0.02 | 0.01 | 19.0 | 1.4 | 7 |
| [20 – 40] | 291 | 0.139 | 0.62 | 1.25 | 0.006 | 0.01 | 0.01 | | | |
| **Aosta** | | | | | | | | | | |
| [0 – 10] | 57 | 0.097 | 0.67 | 2.05 | 0.050 | 0.12 | 0.03 | | | |
| [10 – 20] | 139 | 0.079 | 0.75 | 2.10 | 0.010 | 0.03 | 0.02 | 13.4 | 2.7 | 20 |
| [20 – 40] | 128 | 0.153 | 0.6 | 2.63 | 0.045 | 0.05 | 0.04 | | | |
| **Valencia** | | | | | | | | | | |
| [0 – 10] | 32 | 0.142 | 0.61 | 1.90 | 0.025 | 0.03 | 0.02 | | | |
| [10 – 20] | 193 | 0.127 | 0.64 | 1.91 | 0.007 | 0.01 | 0.01 | 21.1 | 1.6 | 8 |
| [20 – 40] | 374 | 0.152 | 0.62 | 2.20 | 0.007 | 0.01 | 0.01 | | | |

**Table II: for each class and each site are listed the number of data points, the optimal values of calibration constants, their estimated errors, the mean value of $W$ and the estimated uncertainty of $W_P$**





| R$^2$ (N $_{points}$); slope, intercept (mm) | | RMSD (mm); %RMSD | Bias (mm) ; % Bias |
|---|---|---|---|
| Classes (mm) | $W_{GPS2}$, $W_P$ | $W_{GPS2}$, $W_P$ | $W_{GPS2}$ -$W_P$ |
| **Rome** | | | |
| [0 – 10] | 0.88 (162); 0.85,1.22 | 0.75; 9.17 | -0.03; 0.60 |
| [10 – 20] | 0.90 (215); 0.97, 0.60 | 1.11; 8.09 | 0.21; 1.89 |
| [20 – 40] | 0.90 (424); 0.81,4.76 | 1.57; 5.64 | -0.39; -0.88 |
| All classes | 0.98 (722); 0.95, 0.84 | 1.35; 6.43 | -0.20; -0.05 |
| **Aosta** | | | |
| [0 – 10] | 0.86 (191); 0.76,1.26 | 1.29; 18.00 | -0.43; -1.72 |
| [10 – 20] | 0.80 (247); 0.77,2.84 | 2.10; 13.20 | -0.82; -3.70 |
| [20 – 40] | 0.71 (131); 0.62,7.32 | 2.61; 10.89 | -1.72; -6.38 |
| All classes | 0.95 (468); 0.88,0.84 | 1.97; 13.57 | -0.88; -3.45 |
| **Valencia** | | | |
| [0 – 10] | 0.79 (122); 0.72,2.01 | 1.13; 14.51 | -0.17; 0.52 |
| [10 – 20] | 0.79 (372); 0.81,2.74 | 1.58; 9.84 | -0.36; -1.50 |
| [20 – 40] | 0.87 (479); 0.88,3.49 | 1.89; 7.02 | 0.29; 1.57 |
| All classes | 0.96 (877); 0.99,0.23 | 1.67; 8.09 | -0.01; 0.34 |

**Table III: parameters of the statistical analysis in the comparison against sunphotometer and GPS water vapour estimations;**

5 **squared correlation coefficient, slope and intercept of the fitting lines, RMSD and Bias**





| | $R^2$ ( $N_{points}$); slope, intercept (mm) | | RMSD (mm); %RMSD | | Bias (mm) ; % Bias | |
|---|---|---|---|---|---|---|
| Classes (mm) | $W_{AER},W_P$ | $W_{AER},W_{GPS}$ | $W_{AER},W_P$ | $W_{AER},W_{GPS}$ | $W_{AER} - W_P$ | $W_{AER} - W_{GPS}$ |
| **Valencia** | | | | | | |
| [0 – 10] | 0.96(249); 1.00,0.55 | 0.84(78);0.93,0.82 | 0.74; 10.33 | 1.00;14.20 | 0.61; 9.16 | 0.35; 5.76 |
| [10 – 20] | 0.94(800); 0.88, 2.05 | 0.90(247);0.83,2.59 | 0.89; 5.56 | 1.15;7.34 | 0.14; 1.59 | 0.003;-1.02 |
| [20 – 40] | 0.96 (660); 0.90,2.32 | 0.92(119);0.84,3.23 | 0.98; 3.92 | 1.59;6.30 | -0.30; -0.95 | -0.69;-2.28 |
| All classes | 0.99 (1481); 0.94, 1.18 | 0.97(383);0.91,1.29 | 0.91; 4.95 | 1.28;7.62 | 0.01; 1.61 | -0.12;-0.97 |

**Tab IV: parameters of the statistical analysis in the comparison against $W_{AER}$ of $W_P$ and $W_{GPS}$ estimations; squared correlation coefficient, slope and intercept of the fitting lines, RMSD and Bias**

