# Peer review of "Precipitable water vapor content from ESR/SKYNET Sun-sky radiometers: validation against GNSS/GPS and AERONET over three different sites in Europe."

_Atmospheric Measurement Techniques, 2017_

## Referee Comment (RC1) · Anonymous Referee #2 · 11 Oct 2017

**Interactive comments on "Precipitable water vapor content from ESR/SKYNET Sun-sky radiometers: validation against GNSS/GPS and AERONET over three different sites in Europe" by M. Campanelli et al.**

This paper presents a validation of a method used for the retrieval of water vapor column from sun-sky radiometer measurements at 940 nm. This method has been already described and applied by the authors in previous works (Campanelli et al., 2010; 2014). In this study, the estimated water vapor column data for three sites characterized by different atmospheric conditions and pollution loadings are validated against GPS and CIMEL sun photometer datasets.

The paper is well written, the results are sufficiently presented and the comparison is of interest of the community. I recommend the publication after the suggested minor revisions.

**Specific comments and technical corrections**

Page 5, line 5: "Water vapour content in the troposphere affects GNSS signals by lowering their propagation velocities 5 with respect to vacuum." → Please, give a reference.

Page 5, lines 11-14: "Since many years,… in a routinely way." → Please, provide reference(s).

Page 5, line 18: How is the ZTD (Zenith Total Delay) defined? Please, explain what ZTD refers to and give reference.

Page 5, line 31: Define the ARPA acronym.

Page 6, lines 3-4: Again, give a definition for Zenith Hydrostatic Delay (ZHD) and the Zenith Wet Delay (ZWD).

Page 6, line 12: The abbreviation PWV is not used/defined anywhere else in the manuscript. Consider changing it to W, which is used for precipitable water vapor throughout the text.

Page 8, line 20: Again, define the acronym RMSD.

Page 8, line 25: "…within 15 min before and after the sun-sky radiometer measurements…" → Do you mean within 15 minutes around the measurement or 15 minutes before and 15 minutes after the measurement, i.e. within 30 minutes?

Page 10, line 7: typo: lover W → lower W

Page 10, line 12: "…very close to the sea, from where humid airmasses are transported all over the day"; Page 10, line 15: "…due to the presence of a breeze circulation, advecting air from the sea"; Page 10, lines 23-24: "However, further analyses, as the correlation between the humidity and the wind, are necessary to confirm this point." → Are there any data of wind speed and direction available for the three locations? It would be interesting to see the wind patterns and if there is any correlation with the water vapor content.

Page 10, line 17: typo: par → part

Page 10, lines 26-27: "…comparing measurements within 1 minute of difference" → It is not clear to me if you use the closest value within 1 min or you compare averages within 1 min.

Page 11, line 17: "…for measurements within 1 minute" → Again, do you mean averaged within 1 min?

Page 18, Fig. 2: In the bottom panel you should change the symbol for Aosta to open squares.

Page 20, Fig. 4: You should mention in the caption that the measurements presented here refer to summer season. Why not presenting the histograms for all seasons? At least consider including winter measurements. Also, mention in the figure caption that the frequency scales are different for the three histograms.

Pages 22-23, Fig. 6 and 7: Consider mentioning the site in the captions. Especially in Fig 7 where Valencia is not mentioned at all.

Page 27, Table IV: typo: Tab → Table. Again, consider mentioning the measurement site in the caption.

---

## Referee Comment (RC2) · Anonymous Referee #1 · 11 Oct 2017

The present study is reaching a very crucial question about the quality of columnar Water Vapor retrieved from sunphotometers and how it is influenced by the selection of coefficients. The general method has been described in previous works by Campanelli et al. (2014). In this work, datasets from 3 station are validated against GPS and AERONET retrievals. The coefficients were recalculated very frequently and the idea of using different ones according to the values of W, is introduced. The approach adapted in the current paper could be applied to other photometers that could measure direct sun irradiance at water absorbing wavelengths. Thus the work is important for the

scientific community and I suggest to accept this study for publication in Atmospheric Measurement Techniques journal, after minor revisions.

General comments: i) It is very interesting the approach of having the coefficients re-calculated every second day. I think it would be extremely interesting to have some results presented on the day to day variations or stability of them or even some link to other atmospheric variables if possible. At the end, how much is the final W retrieval is affected if a more infrequent schedule is adopted. It would be really valuable to con-clude some guidelines on the operational use of the method on this prospective. ii) All the statistics of the biases among GPS and POM retrievals are presented in a relative approach in the text. Only in table 3 and 4 are presented absolute values of biases. It would be really useful to add some absolute values and corresponding statistics in the discussion. For example figure 5, suggests that the spread of differences should be almost in the same order at all 3 classes and probably lower at the very low W class. There the absolute values of the biases would add more to the interpretation of the intercomparison.

Specific comments

p.1 line 30. RMSD is not a well know abbreviation. It should be written in full form here. Section 2. At the description of the 3 sites I would suggest to add some more info regarding important aspects of the sunphotometric methods, such as statistics about hours of sunlight or cloud coverage or expected SZA range throughout seasons. Also, it should be added that Rome and Aosta are in Italy, as not all readers are not familiar with south Europe. p. 4 line 9 Are there any differences in the 940nm channel between POM 1 and POM 2? If yes, report them and also report which was used in each of the 3 datasets. p.4 section GNSS/GPS. Since GPS receivers are located up to 7km from the sunphotometers, it would be really useful to have some reference on the spatial variability of W, and how much it could affect the validation. p.5 line18 ZTD some def-inition on ZTD is needed. p.5 line 30 NWP abbreviation is not explained anywhere in the text. p.6 line 23. And p.7 line 4 A little attention in explaining T. The formula written

[Figure]

here is only the transmittance due to the presence of W in the atmosphere. The way it is written is seems that there is no dependence on aerosols and Rayleigh scattering in this bandpass. Restate this sentence so that this is clear. p.9 line 19-20. The uncertainty calculated here is just the relative deviation of GPS and POM retrievals. This is a statistical measure which shows very well how the biases are spread. But it is not the total uncertainty of the retrieval which should include instrumental uncertainties, errors introduced at different steps of the method and their spread, and any other systematic errors. It should be restated so that is clear that this is not the total uncertainty of W retrieval. p10- line 16. Although it is not presented somewhere in the study, I assume that higher uncertainties are expected in sunphotometric methods at very high SZAs, which is usually the case in early morning and late afternoon. I suggest adding some information and discussion about that at this point.

Figure 3. the caption is note descriptive enough. It should be restated to be clear what are the data points in this plot.

Figure 7: It is not stated which station's dataset is used in these plots.

Please also note the supplement to this comment:
https://www.atmos-meas-tech-discuss.net/amt-2017-221/amt-2017-221-RC2-supplement.pdf

---

## Author Comment (AC1) · 7 Nov 2017

We thank the reviewer for the positive general comment on our work.
We answered point by point to his comments as shown below:

***Page 5, line 5: "Water vapour content in the troposphere affects GNSS signals by lowering their propagation velocities 5 with respect to vacuum." → Please, give a reference.***

We added the following reference:  Saastamoinen, 1973; Bevis et al., 1992

***Page 5, lines 11-14: "Since many years , ... in a routinely way." → Please, provide reference(s).***

Bennitt and Jupp, 2012; Guerova et al., 2016 has been added.

***Page 5, line 18: How is the ZTD (Zenith Total Delay) defined? Please, explain what ZTD refers to and give reference.***

The text has been changed (in red the changes):" Dry air and water vapour molecules in the troposphere affects GNSS signals by lowering their propagation velocities with respect to vacuum (Saastamoinen, 1973; Bevis et al., 1992). A diminished speed results in a time delay in the signal propagation along the satellite-receiver path, that multiplied by the vacuum speed of light adds an extra-distance to the satellite-receiver geometrical one. It is worth reminding here that the tropospheric delay (the word delay is usually referred to the extra distance and is expressed in meters) due to the dry air and water vapour molecules, is just one out of many other systematic errors affecting GNSS observations, which are to be accounted for in order to achieve sub-centimeter accuracy positions. During GNSS data processing, the contribution of dry air and water vapour to the total delay are separated and estimated in the zenith direction.  This leads to the definition of three delay parameters: ZTD ( Zenith Total Delay), ZHD ( Zenith Hydrostatic Delay), and ZWD ( Zenith Wet Delay), related by the ZTD=ZHD+ZWD ( Bevis et al., 1992; Guenoca et al:, 2016)"

***Page 5, line 31: Define the ARPA acronym.***

Done

***Page 6, lines 3-4: Again, give a definition for Zenith Hydrostatic Delay (ZHD) and the Zenith Wet Delay (ZWD).***

See the answer above

***Page 6, line 12: The abbreviation PWV is not used/defined anywhere else in the manuscript. Consider changing it to W, which is used for precipitable water vapor throughout the text.***

done

***Page 8, line 20: Again, define the acronym RMSD.***

done

*Page 8, line 25: "...within 15 min before and after the sun-sky radiometer measurements..."* → *Do you mean within 15 minutes around the measurement or 15 minutes before and 15 minutes after the measurement, i.e. within 30 minutes?*

The sentence was changed as "The closest $W_{GPS}$ retrievals within 30 minutes, 15 min before and after the sun-sky radiometer measurements were selected."

*Page 10, line 7: typo: lover W → lower W*

done

*Page 10, line 12: "...very close to the sea, from where humid airmasses are transported all over the day"; Page 10, line 15: "...due to the presence of a breeze circulation, advecting air from the sea"; Page 10, lines 23-24: "However, further analyses, as the correlation between the humidity and the wind, are necessary to confirm this point." → Are there any data of wind speed and direction available for the three locations? It would be interesting to see the wind patterns and if there is any correlation with the water vapor content.*

We inserted information of wind direction and the correspondent transported W for all the sites. We modified the text as follows:

 "Looking at Figure 4a, referred to summertime, it is worth highlighting that Valencia is the site where high $W$ values  (>30 mm) are more homogenously distributed over time, with a very slight increment in the afternoon due to breeze circulation. This is principally due to the location of this site, very close to the sea, from where humid air masses are transported all over the day. This kind of distribution of greater water vapor content is visible also in the other seasons, showing a sort of homogeneity of $W$ distribution all over the year. In Figure 4b a bivariate polar plot with smoothing, obtained from openair package, is shown. W content, for the entire year, in polar coordinates is shown by wind speed (radius of the circles) and direction. Mean contents are calculated for wind speed-direction 'bins' (e.g. 0-1, 1-2 m/s,...  and 0-10, 10-20 degrees etc). It is evident from this plot that the largest amount of W is brought from Easterly winds, being the seacoast 10km East from the site.
In Rome $W$ values >35 mm are mostly recognizable during summer afternoons, from about 14 UTC, due to the presence of a breeze circulation, advecting air from the sea (Figure 4c). The importance of wind from SW (that is from the sea) in transporting $W$ to the site, is highlighted in Figure 4d, whereas lower $W$ content is mostly recorded when wind comes from N direction, having also the highest speed.   In all seasons greater water vapor content is retrieved in the early morning and late afternoon showing, also for this site, a generally homogeneous $W$ yearly distribution. A smaller number of measurements is available in Rome during the middle part of the day in all seasons. This is mostly due to the formation of convective clouds at around 12 UTC, favored by the urban heat island phenomenon, that didn't allow the photometer to operate.
In Aosta, as shown in Figure 4f, the greater amount of $W$ comes from East direction, that is from the Po Valley, a humid region with higher atmospheric stability and weaker winds, and mostly during summer and autumn seasons; elevated values of $W$ (>35 mm) during summer

were retrieved more frequently in the morning, but this hourly distribution was found also in autumn for *W*>25 mm (Figure 4e). This behavior could be caused by the atmospheric stability; in the late morning, especially in summer and fall when the insolation is higher, valley-mountain flows develop mixing the humid air of the lower levels with the dried air above. Then, winds aloft could remove part of this humidity by advection, decreasing the water content of the air column. The other seasons conversely show more homogeneous *W* distribution during the day. Low W content associated to winds from W, is due to the Foehn. When wind comes from this side, air masses passed the Alps and arrived over Aosta drier."

[Figure]

**Figure 4: plots a, c, e, -polar plot showing the distribution of W values, during summer season, grouped according to their numeric range; the 24 quadrants are hours in UTC; the radius represents the frequency of events normalized to the number of point of the season. The frequency scales are different for the three histograms. Plots b, d, f,- bivariate polar plot with smoothing, showing the distribution of W content, for the entire year, by wind speed (radius of the circles) and direction.**

*Page 10, line 17: typo: par → part*

Done

*Page 10, lines 26-27: "...comparing measurements within 1 minute of difference" → It is not clear to me if you use the closest value within 1 min or you compare averages within 1 min.*

The sentence has been changed "within 1 minute of difference (if more than one measurement of $W_{GPS2}$ was found, their average was performed)"

*Page 11, line 17: "...for measurements within 1 minute" → Again, do you mean averaged within 1 min?*

The sentence has been changed "within 1 minute of difference (if more than one measurement of $W_{AER}$ was found, their average was performed)"

*Page 18, Fig. 2: In the bottom panel you should change the symbol for Aosta to open squares.*

done

*Page 20, Fig. 4: You should mention in the caption that the measurements presented here refer to summer season. Why not presenting the histograms for all seasons? At least consider including winter measurements. Also, mention in the figure caption that the frequency scales are different for the three histograms.*

The caption was changed as:" polar plot showing the distribution of W values, during summer season, grouped according to their numeric range; the 24 quadrants are hours in UTC; the radius represents the frequency of events normalized to the number of point of the season. The frequency scales are different for the three histograms. "

Comparing the histograms in the different seasons, we noticed that the information content provided by the winter is not very significant to highlight the main differences among the 3 sites. During this season a smaller number of measurements is available, mostly in Aosta (due also to the sun being behind the mountains) and in Rome (high clouds presence in the middle of the day). Moreover, the range of values assumed by W is not very wide (see below Figures). Spring and autumn are very similar to summer season, except for the values assumed by W. For these reason we considered not important adding more plots.

[Figure]

*Pages 22-23, Fig. 6 and 7: Consider mentioning the site in the captions. Especially in Fig 7 where Valencia is not mentioned at all.*

done

*Page 27, Table IV: typo: Tab → Table. Again, consider mentioning the measurement site in the caption.*

done

---

## Author Comment (AC2) · 7 Nov 2017

We thank the reviewer for the positive general comment on our work.
We answered point by point to his comments as shown below:

*General comments:*
*i) It is very interesting the approach of having the coefficients recalculated every second day. I think it would be extremely interesting to have some results presented on the day to day variations or stability of them or even some link to other atmospheric variables if possible. At the end, how much is the final W retrieval is affected if a more infrequent schedule is adopted. It would be really valuable to conclude some guidelines on the operational use of the method on this prospective.*

We agree with the suggestion of the reviewer. A sensitivity study about the time frequency of external measurements (both daily and monthly but also inside a single day) is needed for building a guideline on the operational use of the method, maybe also delivering a software. However, we believe that such study needs time and an accurate analysis, and it could take to an addition of more sections to this paper, already long.
Therefore, we retain this idea very good for further investigations and we explicitly mentioned it as a future prospect of the research.
We added the following sentence at the end of the conclusions:
Finally, a sensitivity study about the time frequency of the independent external measurements (both daily and monthly but also inside a single day) will be a future prospect of this research in order to build a guideline on the operational use of the methodology and delivering a software

*ii) All the statistics of the biases among GPS and POM retrievals are presented in a relative approach in the text. Only in table 3 and 4 are presented absolute values of biases. It would be really useful to add some absolute values and corresponding statistics in the discussion. For example figure 5, suggests that the spread of differences should be almost in the same order at all 3 classes and probably lower at the very low W class. There the absolute values of the biases would add more to the interpretation of the intercomparison.*

Absolute values of biases for each class of Figure 5 are already listed in table III. We commented the results in terms of absolute values as below ( red are new comments):

"The comparison between $W_P$ and $W_{GPS2}$ for Rome and Valencia (Table III) when all $W$ classes are analysed, shows high $R^2$, varying from 0.98 and 0.96; RMSD assumes values from 1.35 mm (6.43%) and 1.67 mm (8.09%), and the Bias is within
-0.01 mm (0.34%) and -0.20 mm (-0.05%), therefore within the estimated error $\Delta W_P$. Investigating separately the 3 classes (divided using the thresholds on the $W_{GPS2}$ dataset) the greatest difference was found for the first class in terms of %RMSD (9.17% - 14.51%) but for the same class it was the smallest in terms of absolute RMSD (0.75 mm – 1.13 mm); conversely for the same class the smallest difference was found in terms of Bias, both in percentage and absolute values, varying from
-0.03 mm (0.60%) and -0.17 mm (-0.52%). However, each class remained within the $\Delta W_P$ error.
The retrieval of $W_P$ for Aosta was generally less performing than for the other sites. For the entire $W$ classes, RMSD and Bias were found to be the highest values, being 13.57% (1.97

mm) and -3.45%, (-0.88 mm) respectively, while $R^2$ is the lowest among the three sites (0.95). Also for this site the greatest value of %RMSD (18.00%) and the smaller one of RMSD (1.29 mm) was found for the first class and the Bias (both percentage and absolute values) remained for each class within the $\Delta W_P$ error. The lower quality performance of the methodology in this site is discussed in section 4. "

*Specific comments*
*p.1 line 30. RMSD is not a well know abbreviation. It should be written in full form here.*

Done

*Section 2. At the description of the 3 sites I would suggest to add some more info regarding important aspects of the sunphotometric methods, such as statistics about hours of sunlight or cloud coverage or expected SZA range throughout seasons. Also, it should be added that Rome and Aosta are in Italy, as not all readers are not familiar with south Europe.*

The location of all the site is already declared at the beginning of the section 2, after the coordinates.
We added the following information at the end of each paragraph describing the sites :
Zenith angle for Rome varies within the interval [18.46° - 65.31°], and hours of sunlight recorded in 2010 were 2431.8 8 (provided by the Italian air Force)..
Zenith angle for Burijassot varies within the interval [16.07° - 62.91°] and hours of sunlight recorded in 2011 were 2678.7.
Zenith angle for Aosta varies within the interval [22.31° - 69.14°] and hours of sunlight recorded in 2011 were 2396.

**p. 4 line 9 Are there any differences in the 940nm channel between POM 1 and POM 2? If yes, report them and also report which was used in each of the 3 datasets.**

The 940 nm filters installed by PREDE in both the models is the same :  Fujitok  with 50%BW = 10 nm.

**p.4 section GNSS/GPS. Since GPS receivers are located up to 7km from the sunphotometers, it would be really useful to have some reference on the spatial variability of W, and how much it could affect the validation.**

Unfortunately there isn't any other historical measurement of W around the 3 sites, useful for analysing the spatial variability of W. However for Rome (2 km distant) and Aosta ( 0.5 km) the GPS is practically collocated. The only site where the antenna is 7 km far from the sun-photometers location is Valencia, but due to the orography of this site we retain W distribution homogeneous.

**p.5 line18 ZTD some definition on ZTD is needed.**

Done

**p.5 line 30 NWP abbreviation is not explained anywhere in the text.**

Done

**p.6 line 23. And p.7 line 4**
**A little attention in explaining T. The formula written here is only the transmittance due to the presence of W in the atmosphere. The way it is written is seems that there is no dependence on aerosols and Rayleigh scattering in this bandpass. Restate this sentence so that this is clear.**

We restated the sentence as below:

Precipitable water vapour content from ESR/PREDE-POM sun-sky radiometer ($W_P$) was calculated using the methodology described in Campanelli et al. (2014). For specific spectral regions in the near infrared, where absorption of dominant trace gases can be considered negligible, we can express the transmittance of the atmosphere ($T_{atm}$) as follows: $T_{atm} = e^{-m_0(\ \tau_R + \tau_a)} \cdot T$ , where $m_0$ is the relative optical airmass (Kasten and Young, 1989), $\tau_a$ and $\tau_R$ are the extinction aerosol optical depth and the molecular Rayleigh scattering at 940 nm respectively, and $T$ is the transmittance of the water vapour, $T = e^{-a(mW)^b}$ with $m$ the water vapour optical airmass, calculated according to Kasten, 1966, and $W$ the columnar water vapour content (Bruegge et al., 1992).

**p.9 line 19-20. The uncertainty calculated here is just the relative deviation of GPS and POM retrievals. This is a statistical measure which shows very well how the biases are spread. But it is not the total uncertainty of the retrieval which should include instrumental uncertainties, errors introduced at different steps of the method and their spread, and any other systematic errors. It should be restated so that is clear that this is not the total uncertainty of W retrieval.**

The following sentence has been added after formula 7:

It must be beard in mind that this uncertainty is a statistical measure but not the total uncertainty of W retrieval which should include instrumental uncertainties, errors introduced at different steps of the method and their spread, and any other systematic errors.

**p10- line 16. Although it is not presented somewhere in the study, I assume that higher uncertainties are expected in sunphotometric methods at very high SZAs, which is usually the case in early morning and late afternoon. I suggest adding some information and discussion about that at this point.**

We missed to write an information about the data selection in term of airmass value. We added the following sentence at the beginning of Section 4. In order to limit the influence of largest uncertainties at very high solar zenith angles, we selected the data having m<8.
However a sensitivity study about the uncertainty introduced for high SZA can be added to study about sensitivity to the time frequency of external independent measurements.

**Figure 3. the caption is note descriptive enough. It should be restated to be clear what are the data points in this plot.**

The caption has been changed as below:

Temporal behaviors of $W_P$ retrieved with the presented methodology, for the years 2010 (Rome), 2011 (Valencia) and 2014 (Aosta).

**Figure 7: It is not stated which station's dataset is used in these plots.**
Done